# Optimization of Cyclohexanol and Cyclohexanone Yield in the Photocatalytic Oxofunctionalization of Cyclohexane over Degussa P-25 under Visible Light

**DOI:** 10.3390/molecules24122244

**Published:** 2019-06-15

**Authors:** Adolfo Henríquez, Victoria Melin, Nataly Moreno, Héctor D. Mansilla, David Contreras

**Affiliations:** 1Laboratorio de Recursos Renovables, Centro de Biotecnología, Universidad de Concepción, 40730386 Concepción, Chile; adohenriquez@udec.cl (A.H.); victoriamelin@udec.cl (V.M.); natalymoreno@udec.cl (N.M.); 2Faculty of Chemical Sciences, Universidad de Concepción, 4070371 Concepción, Chile; hmansill@udec.cl; 3Millennium Nuclei on Catalytic Processes towards Sustainable Chemistry, 7810000 Santiago, Chile

**Keywords:** experimental design, optimization, photocatalysis, oxofunctionalization, cyclohexane, TiO_2_, Degussa P-25

## Abstract

The sustainable transformation of basic chemicals into organic compounds of industrial interest using mild oxidation processes has proved to be challenging. The production of cyclohexanol and cyclohexanone from cyclohexane is of interest to the nylon manufacturing industry. However, the industrial oxidation of cyclohexane is inefficient. Heterogeneous photocatalysis represents an alternative way to synthesize these products, but the optimization of this process is difficult. In this work, the yields of photocatalytic cyclohexane conversion using Degussa P-25 under visible light were optimized. To improve cyclohexanol production, acetonitrile was used as an inert photocatalytic solvent. Experiments showed that the use of the optimized conditions under solar light radiation did not affect the cyclohexanol/cyclohexanone ratio. In addition, the main radical intermediary produced in the reaction was detected by the electronic paramagnetic resonance technique.

## 1. Introduction

Many different chemical compounds are required to maintain the lifestyles that humans have become accustomed to. The present state-of-the-art processes for synthesizing chemical products are highly inefficient because of the high energy that is required and/or the large amount of chemical waste that is generated [1]. Two examples of such products are cyclohexanol and cyclohexanone, which are oxygenated compounds produced by the oxidation of cyclohexane. These oxidation products are very important in the chemical industry for their use in the manufacture of adipic acid, caprolactam, or commodities such as nylon-6 and nylon-66 [2]. The industrial oxidation of cyclohexane is usually performed in conditions that include high energy consumption, temperatures between 170 and 230 °C, and pressure of over 1 MPa. In addition, the conversion rate is controlled so that it remains below 10% to minimize undesired oxidation reactions and CO_2_ formation [3]. In the search for alternative pathways to produce chemical precursors, the use of stoichiometric oxidants and catalysts has been studied. In this context, heterogeneous catalysts and molecular oxygen as an oxidant have received attention in the last several years because heterogeneous semiconductor materials allow for chemical conversions using visible light irradiation as an energy source under ambient pressure and temperature conditions [4,5]. Moreover, the development of processes that use a semiconductor with photocatalytic activity in the range of solar radiation is particularly relevant to realizing innovative and economically advantageous processes that convert hydrocarbons and, at the same time, move toward a “sustainable chemistry” that has minimal environmental impact [6].

Titanium dioxide is the most widely used semiconductor photocatalyst because it is highly photoreactive, biologically and chemically inert, inexpensive, photostable, and nontoxic [7]. This photocatalyst has a large band gap energy (3.0 eV for rutile and 3.2 eV for anatase), and thus, it does not absorb visible light [8]. Although there are many different sources of titanium dioxide, Degussa P-25 TiO_2_ has effectively become the research standard because of its well-defined nature [9]. Titanium dioxide Degussa P-25 powder is a standard material with high photocatalytic activity. It consists of anatase and rutile phases (70:30) and has a relatively large specific area (49 m^2^ g^−1^) [10]. This photocatalyst has been studied in several areas, such as environmental remediation [11], energy production [12], and benign organic synthesis [13]. In this context, the use of titanium dioxide Degussa P-25 represents an interesting alternative for the synthesis of oxygenated compounds from cyclohexane. The yields and selectivity of oxidation products are variable and related to the reaction conditions. Although increasing the yield is an important goal to achieve an economically sustainable process, the main challenge is improving the cyclohexanol selectivity since this product is overoxidized, resulting in the preferential formation of cyclohexanone. The structural modification of TiO_2_ [14,15], the addition of cocatalysts to the reaction systems [16,17], or the use of solvents [18,19] are some of the alternative methods that lead to the oxidation of cyclohexane. The adsorption and desorption process of cyclohexane and their oxidation products over TiO_2_ surface [20,21] are solvent-dependent because the polar properties of TiO_2_ and the apolar properties of cyclohexane can be successfully employed to improve the yields of cyclohexanol. For this purpose, the solvent must be inert and stable under photocatalytic conditions [22]. 

In this work, the selective oxofunctionalization of cyclohexane to cyclohexanol and cyclohexanone by titanium dioxide Degussa P-25 was optimized under simulated solar light using acetonitrile as a solvent to generate oxidation conditions that increase the yield of cyclohexanol. Additionally, the radical intermediate produced on the surface of the catalyst was identified.

## 2. Results and Discussion

### 2.1. TiO_2_ Degussa P-25 Characterization

To assess the physical properties of the Degussa P-25 catalyst, the commercial photocatalyst was characterized (Figure 1). Figure 1a shows the spectrum of diffuse reflectance; as shown by the inset of the Kubelka–Munk plot, an energy band of 3.2 eV was confirmed. The adsorption–desorption isotherm of N_2_ corresponds to an isotherm of type II, indicating that this material does not have porosity and has a specific area of 49 m^2^ g^−1^ (Figure 1b). According to the literature, the size of these particles is 20–30 nm for anatase particles and 80–90 nm for rutile particles [23]. 

### 2.2. Effect of Acetonitrile on the Yield of Cyclohexane Conversion

A preliminary experiment was performed to determine the effect of acetonitrile on the conversion yield of cyclohexane to cyclohexanol and cyclohexanone. This experiment was performed with 1 g L^−1^ of TiO_2_ (Figure 2), which is a standard concentration of this catalyst according to the literature [24,25,26]. Two photocatalysis reactions were carried out (1) using only cyclohexane as the solvent and (2) using acetonitrile as the solvent with 2% cyclohexane. In the system with acetonitrile, cyclohexanone conversion was 15 times higher than the conversion in pure cyclohexane. The conversion of cyclohexanol was 380 times higher when acetonitrile was added to the reaction system. These results suggest that the reaction develops more efficiently in a polar solvent such as acetonitrile. On the other hand, the selectivity of the oxofunctionalization reaction increased significantly toward cyclohexanol when it was carried out in acetonitrile. This can be explained by the competitive adsorption between the oxidation products and the solvent molecules on the photocatalyst surface.

### 2.3. Optimization of Cyclohexane Oxofunctionalization

The effect of water on the reactivity of the TiO_2_ photocatalyst is the result of the oxygen molecule in adsorbed water serving as a center for radical species generation. In addition, the water on the surface of the catalyst modifies the adsorption properties of the TiO_2_ catalyst for the substrate, intermediates, and products. On the other hand, the photocatalytic oxidation of cyclohexane on TiO_2_ cannot be performed in water because cyclohexane is apolar. In preliminary photocatalytic experiments, aqueous microemulsions of cyclohexane on TiO_2_ were treated (results not shown). In these systems, oxidation products of cyclohexane were not detected. The hypothesized mechanism that underlies this phenomenon is that the catalyst surface is saturated with water, preventing adsorption of cyclohexane on the catalyst. Different behavior was observed without water, as shown in the previous section. Taking into account the higher yield of cyclohexane oxidation in acetonitrile and the importance of the adsorbed water on the surface of TiO_2_, the dependence of cyclohexanol yield and cyclohexanone yield (dependent variables) on the water concentration, and amount of TiO_2_ (independent variables) was determined by multiple linear regression (MLR) from multivariate experiments. In addition, the optimal experimental conditions were determined. 

From MLR analysis, two validated polynomials were obtained (Equations (1) and (2)).
(1)$$Y\left(\mathrm{CHol}\right)=591\left(\pm 17\right)-70\left(\pm 12\right)\;\mathrm{cat}+10\left(\pm 12\right)\;\mathrm{water}-118\left(\pm 12\right){\;\mathrm{cat}}^{2}-110\;\left(\pm 12\right){\;\mathrm{water}}^{2}+32\left(\pm 19\right)\;\mathrm{cat}*\mathrm{water}$$
(2)$$Y\left(\mathrm{CHone}\right)=1330\left(\pm 32\right)-127\left(\pm 22\right)\;\mathrm{cat}+7\left(\pm 22\right)\;\mathrm{water}-279\left(\pm 24\right){\;\mathrm{cat}}^{2}-242\;\left(\pm 24\right){\mathrm{water}}^{2}+49\left(\pm 35\right)\;\mathrm{cat}*\mathrm{water}$$

In these models, the independent variables were the amount of catalyst and water in the 2% *v/v* solution of cyclohexane (185 mmol L^−1^) in acetonitrile, and the dependent variables were yield of cyclohexanol (Y_CHol_) and cyclohexanone (Y_CHone_). 

The 3-D graphical representations of these polynomials are shown as response surfaces in Figure 3a,b. From these polynomials and surface response plots, the optimal concentrations of TiO_2_ (5.3 g L^−1^) and water (2400 mmol L^−1^) were determined. In these conditions, the activity of the catalyst was optimal because the surface of TiO_2_ must be hydrated to be reactive for reactive oxygen species (ROS) production. In addition, the optimal amount of catalyst is related to the light dispersion at higher values. The synergic interaction between the amount of the catalyst and water was significant, and this is shown by the coefficients of the interaction terms in Equations (1) and (2). It is highlighted that the yield of cyclohexanone is twice the yield of cyclohexanol.

### 2.4. Cyclohexane Oxofunctionalization under Sunlight Irradiation

Next, the effect of the irradiation source on the cyclohexanol and cyclohexanone yields obtained from the photocatalytic oxofunctionalization reaction of cyclohexane catalyzed by TiO_2_ Degussa P-25 was evaluated. To this end, experiments under a metal halide lamp and real solar radiation were performed at the optimal conditions determined by the experimental design.

In these experiments, higher yields of cyclohexanol and cyclohexanone were obtained when the metal halide lamp was used as the irradiation source (Figure 4). The cyclohexanol/cyclohexanone ratio observed under the metal halide lamp was 0.554, while the ratio observed under solar light was 0.566. 

### 2.5. Radical Intermediates Detected by Electron Paramagnetic Resonance (EPR) Spectroscopy

In order to identify the radicals generated during the oxofunctionalization of cyclohexane in optimal experimental conditions, a spin trap EPR experiment was performed using 5,5-dimethyl-1-pyrroline *N*-oxide (DMPO) and *N*-tertbutyl-α-phenylnitrone (PBN). The EPR spectrum of radical–DMPO (Figure 5a) corresponds to a triplet of doublet signals (a_N_ = 13.0 G; a_H_ = 6.0 G). On the other hand, the EPR spectrum of radical–PBN (Figure 5b) corresponds to a broad triplet signal (a_N_ = 13.6 G). A comparison of the hyperfine coupling constants of the above EPR spectra with literature values for oxygen-centered radicals indicates that the adduct detected in both cases is formed by the spin trap and the C_6_H_11_O• radical [27]. However, it has been demonstrated that the trapping of organic peroxyl radicals at room temperature produces alkoxyl rather than peroxyl radical adducts. In this way, C_6_H_11_O•, C_6_H_11_OO•, or both species are intermediates in the reaction of cyclohexane oxidation over TiO_2_ [27]. It is highlighted that •OH radicals were not detected in the EPR experiments. However, as the amount of water is low and the •OH radical is highly reactive, the production of this ROS cannot be excluded from our reaction system. It is most likely that the •OH radicals produced remain bonded to the surface of the photocatalyst, as some authors have suggested [28,29]. 

Conte et al. [30] reported that the first step in the oxidation of hydrocarbons over a TiO_2_-based photocatalyst is hydrogen abstraction, which produces an acyl radical. This suggests that the first step in the photocatalytic oxidation of cyclohexane over Degussa P-25 is the abstraction of hydrogen from cyclohexane by a photo-hole of the photocatalyst (Equation (3)). 

(3)$$C_{6}H_{12}+h^{+}\to C_{6}H_{11}{}^{•}+H^{+}$$

However, this radical could also be produced by the •OH radical attack (Equation (4)).
(4)$$C_{6}H_{12}+Ti^{\mathrm{IV}}-\mathrm{OH}•\;\to C_{6}H_{11}{}^{•}+H^{+}$$

Under air-saturated conditions, these radicals react with oxygen to produce cyclohexylperoxy radicals (Equation (5)).
(5)$$C_{6}H_{11}{}^{•}+O_{2}\to C_{6}H_{11}{\mathrm{OO}}^{•}$$

This latter radical, through a disproportionation reaction, produces cyclohexanol and cyclohexanone (Equation (6)).
(6)$$C_{6}H_{11}{\mathrm{OO}}^{•}\to C_{6}H_{11}\mathrm{OH}+C_{6}H_{10}O+O_{2}$$

This reaction pathway has been proposed in the literature for the oxidation of cyclohexane by TiO_2_-based photocatalysts [15,27] and is likely to be the main pathway for the production of cyclohexanol and cyclohexanone from the photocatalytic oxidation of cyclohexane.

## 3. Materials and Methods

### 3.1. Reagents

Titanium dioxide Degussa P-25, cyclohexane, acetonitrile, and the analytical standards cyclohexanol and cyclohexanone were purchased from Sigma-Aldrich (St. Louis, MO, USA). All reagents and solvents were used as received without further purification.

### 3.2. General Procedures

The methodology used in this work and described in Section 3.3, Section 3.4, Section 3.6, and Section 3.8 was performed according Henríquez et al. [30].

### 3.3. Titanium Dioxide Degussa P-25 Characterization

X-ray diffraction (XRD) patterns of titanium dioxide Degussa P-25 were collected using a Bruker D8 Advance X-ray diffractometer (Billerica, MA, USA) using Cu K_α_ radiation as the source with a working voltage and current of 40 kV and 30 mA, respectively. The intensity of the diffraction peaks was recorded in the 10–80° (2θ) range with increments of 0.05° and a counting time of 0.5 s^−1^ per step. 

The UV–Vis spectrum of Degussa P-25 was recorded with an Agilent Cary 5000 UV/Vis/NIR spectrophotometer (Santa Clara, CA, USA). The band gap (E_g_) of the Degussa P-25 photocatalyst was determined from Tauc plots obtained from their respective UV–vis diffuse reflectance spectra. The relational expression proposed by Tauc, Davis, and Mott [31,32] (Equation (7)) was used.
(7)hνα1n=A(hν−Eg)
where h is the Planck’s constant, ν is the frequency of vibration, α is the absorption coefficient, E_g_ is the band gap, and A is a proportional constant. The value of the exponent n denotes the nature of the simple transition. For an indirect allowed transition, n = 2. In the Tauc equation, α is substituted by the Kubelka–Munk function F(R_∞_). The crystal morphology of the products was observed by scanning electron microscopy (FEI–Nova NanoSEM 200, Hillsboro, OR, USA). Nitrogen adsorption isotherms at 77 K were obtained using a BELSORP-mini II surface area and a pore size analyzer (Osaka, Japan).

### 3.4. Cyclohexane Oxofunctionalization

The oxofunctionalization experiments were performed in acetonitrile containing 185 mmol L^−1^ cyclohexane. The reaction was performed in a 50 mL 2-neck round-bottom flask fitted with a reflux condenser. The amount of titanium dioxide Degussa P-25 and the concentration of water were changed according to the multifactorial design. In every case, the system was saturated with 1 atm of air during the reaction time (3 h). Irradiation was carried out under visible radiation, which was generated by a 400 W metal halide lamp (Osram Powerstar HQI-E 400 W/D Pro Daylight, Munich, Germany). A photo synthetically active radiation (PAR) radiometer (QSPL-2100, San Diego, CA, USA), sensitive between 400 and 700 nm was keeping inside reactors arrangement. A picture of this is showed in Appendix A.

### 3.5. Multivariate Experiments

To obtain the optimum yield of cyclohexanol from the photocatalytic oxofunctionalization of cyclohexane in the experimental conditions described in 3.3, a multifactorial experimental design was developed on the basis of a circumscribed composite core design (CCC), which was composed of a two-level factorial design (with levels of −1 and +1), star points (with levels of −√2 and +√2), and replicas of the central point (Table 1). The experimental factors selected as variables were the water concentration (2230–2670 mmol L^−1^) and the amount of catalyst (4.88–6.16 g L^−1^). The range of water concentration and the amount of catalyst were determined according to preliminary studies of the maximum inclination curve. The yield of cyclohexanol and cyclohexanone (μmol) after 180 min of irradiation with metal halide irradiation was selected as the response. The polynomials related to the reaction system and the response surface plot were obtained by analyzing the data with the software Modde 7.0 (Umetrics, Sweden). The model was validated statistically with the same software using analysis of variance (ANOVA) with a confidence level of 90%.

### 3.6. Identification of Oxofunctionalized Products

In order to identify and quantify the compounds obtained from the photocatalytic oxidation of cyclohexane, 1.0 µL aliquots of both the standard solutions and reaction mixture were injected into the GC–MS system. GC–MS analyses were performed in a 5890 Series II gas chromatograph (Hewlett Packard Corporation, Palo Alto, CA, USA) interfaced with a 5972 Mass Selective Detector Quadrupole (Hewlett Packard Corporation, Palo Alto, CA, USA). Analytes were separated in a cross-linked mixture of 5% diphenyl–95% dimethylsiloxane (Hewlett Packard Corporation, Palo Alto, CA, USA). The temperature program was an isotherm at 40 °C for 15 min. Helium was used as the gas carrier at a constant flow of 1 mL/min. Data acquisition was performed in electron impact ionization (EI) and selected-ion monitoring (SIM) modes. The temperature inlet was set at 250 °C, and the source was set at 280 °C.

### 3.7. Solar Experiments

Photosynthetically active irradiance during the course of the experiments performed under metal halide lamp and solar light was measured and recorded using a QSPL-2100 radiometer (Biospherical Instruments Inc., San Diego, CA, USA) equipped with a Teflon^®^ spherical irradiance collector with a 1.9 cm diameter (San Diego, CA, USA). The experiments under real solar radiation were performed on 9 February 2017, between 14:33 and 17:33 h in Concepción city (36°50′00.0″ S, 73°01′49.1″ W). The reactor picture is showed in Appendix A.

### 3.8. In Situ Electron Paramagnetic Resonance Experiments

To evaluate the radical species formed during the cyclohexane oxofunctionalization over the photoactivated surface of Degussa P-25, in situ electron paramagnetic resonance (EPR) measurements were performed with an EMX micro 6/1 Bruker ESR spectrometer working in the X-band and equipped with a Bruker Super High QE cavity resonator (Billerica, MA, USA). *N*-Tertbutyl-α-phenylnitrone (PBN) and 5,5-dimethyl-1-pyrroline *N*-oxide (DMPO) were used as spin traps. First, 1 mg of Degussa P-25 TiO_2_ was dispersed in 1 mL of cyclohexane containing 10 mmol L^−1^ spin trap and 0.1% (*v/v*) nanopure water. Then, the reaction was initiated by turning on the irradiation source. The reaction system was saturated with air during the reaction time. The reactions were carried out in an EPR sample tube (ER 221TUB/, 4 mm I.D.) inside the EPR cavity, irradiated by the 400 W metal halide lamp described above. The in-situ measurements were performed at room temperature. Typical instrumental conditions were as follows: Center field, 3514 G; sweep width, 200 G; microwave power, 20 dB; modulation frequency, 100 kHz; time constant, 0.01 ms; sweep time, 30 s; modulation amplitude, 1.00 G; and receiver gain, 30 dB.

## 4. Conclusions

In this study, the optimal conditions for the photocatalytic oxidation of cyclohexane under visible light were determined. Under these optimized conditions and ambient temperature and pressure, cyclohexane was converted to cyclohexanol and cyclohexanone by heterogeneous photocatalysis using direct solar light as the irradiation source. Although the percentage of cyclohexane conversion by photocatalytic oxidation in this work was lower than that obtained in other works, the system presented in this paper is a cheaper and cleaner alternative for the photocatalytic conversion of cyclohexane. 

The cyclohexanol/cyclohexanone ratio increased when the conversion occurred in a polar environment. Water on the surface of the TiO_2_ catalyst affects the reactivity, and this influence can be attributed to the need for a hydrated catalyst in order to oxidize the substrate by promoting the formation of the •OH radical.

Although the •OH radical was not detected in this work, hydrogen abstraction by the catalyst was detected indirectly since it is required to form C_6_H_11_• as a preliminary step to the formation of C_6_H_11_OO• (Equation (5)). The selectivity of TiO_2_ for cyclohexanone production could be related to the presence of the OH group on the catalyst surface that binds the cyclohexanol formed and oxidizes it to produce cyclohexanone. The selectivity for cyclohexanol or cyclohexanone is difficult to modulate with this catalyst because of its polar nature.

## Figures and Tables

**Figure 1 molecules-24-02244-f001:**
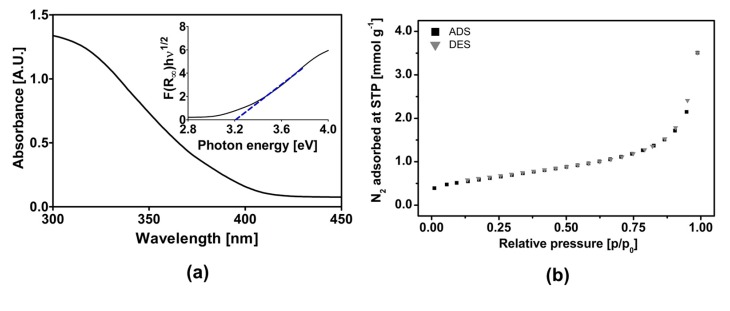
Characterization of commercial photocatalyst Degussa P-25. (**a**). Diffuse reflectance spectrum with Kubelka–Munk graph inset. (**b**). Nitrogen adsorption–desorption isotherms at 77 K.

**Figure 2 molecules-24-02244-f002:**
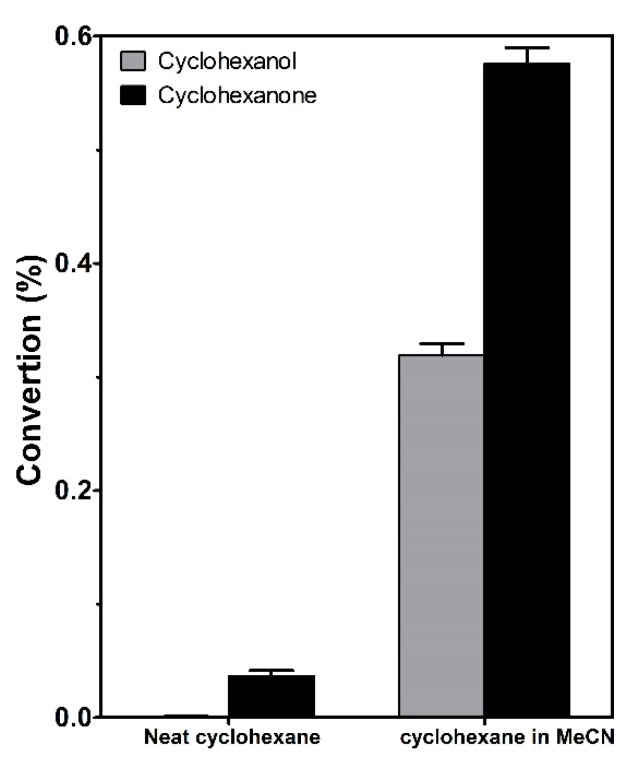
Effect of the solvent on the conversion and selectivity conversion of the photocatalytic oxofunctionalization of cyclohexane using titanium dioxide Degussa P-25 as a photocatalyst (1 g L^−1^).

**Figure 3 molecules-24-02244-f003:**
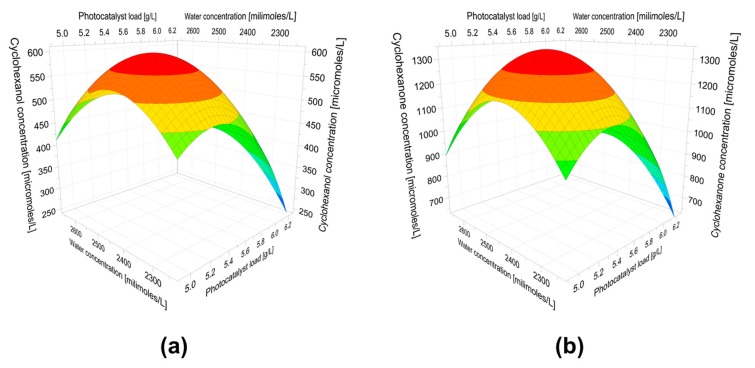
Response surface of the photocatalytic oxofunctionalization of cyclohexane after 180 min with titanium dioxide Degussa P-25 under irradiation of visible light. (**a**) Cyclohexanol optimization, (**b**) Cyclohexanone optimization.

**Figure 4 molecules-24-02244-f004:**
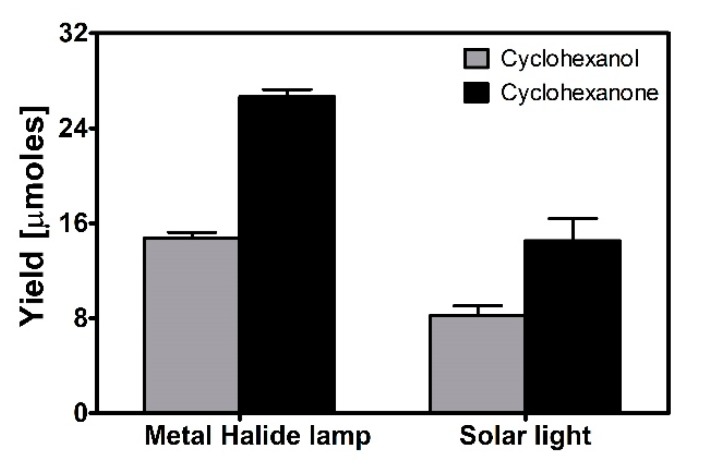
Effect of the irradiation source on the yields of cyclohexanol and cyclohexanone obtained from the photocatalytic oxofunctionalization reaction of cyclohexane catalyzed by TiO_2_ under the optimal conditions determined by experimental design.

**Figure 5 molecules-24-02244-f005:**
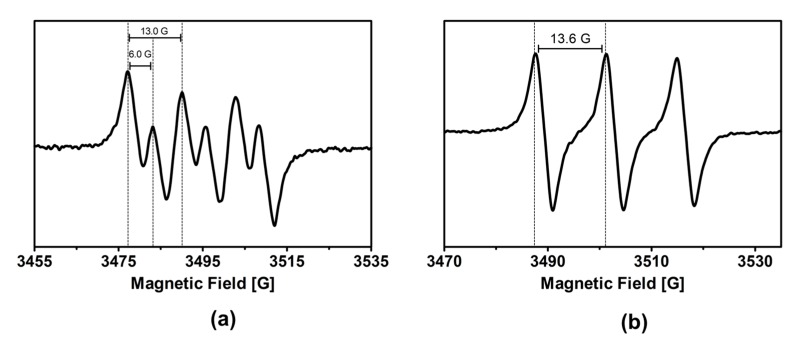
The electron paramagnetic resonance (EPR) spectra obtained 650 s after the photocatalytic oxofunctionalization of cyclohexane saturated with air under visible light irradiation by a 400 W metallic halide lamp in the presence of (**a**) 5,5-dimethyl-1-pyrroline or (**b**) *N*-oxide phenyl-tert-butyl nitrone.

**Table 1 molecules-24-02244-t001:** Experimental design used for the optimization of cyclohexane oxofunctionalization by Degussa P-25 under visible light.

Experiment	Amount of Photocatalyst (g L^−1^)	Concentration of Water (mmol L^−1^)
	4.88 (−1)	2226 (−1)
2	6.16 (1)	2226 (−1)
3	4.88 (−1)	2666 (1)
4	6.16 (1)	2666 (1)
5	4.62 (−√2)	2446 (0)
6	6.42 (√2)	2446 (0)
7	5.52 (0)	2135 (−√2)
8	5.52 (0)	2757 (√2)
9	5.52 (0)	2446 (0)
10	5.52 (0)	2446 (0)
11	5.52 (0)	2446 (0)

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
