# Peer review of "Optimization of Cyclohexanol and Cyclohexanone Yield in the Photocatalytic Oxofunctionalization of Cyclohexane over Degussa P-25 under Visible Light"

_molecules, 2019, doi:10.3390/molecules24122244_

Round 1

Reviewer 1 Report

First of all, the manuscript needs very careful English revision. It contains many grammatical and typographical errors that with no doubt have to be corrected. There are, indeed, sentences where the phrasing is so confusing that the reader may come to false conclusions. In the current form the paper is not acceptable for publication.

Units in the whole manuscript should be standardized. For example, once there is K (line 81), another time °C (line 189).

I believe that in line 36 there should be “MPa“ instead of “mPa".

In line 127 there should be “cyclohexanone” instead of “cyclohexane”.

In line 186 in “Eg” letter g should be written with a lower index.

In lines 203-204 it is said that the ranges of water concentration and amount of catalyst are 2226-2666 mmol/L and 4.88-6.16 g/L, respectively. Meanwhile, in table 1 these ranges are greater (for example experiments 2 and 7). Why?

The figures are wrongly numbered and consequently wrongly mentioned in the text. Additionally there are two “Figures 3” in the manuscript.

Author Response

First of all, the manuscript needs very careful English revision. It contains many grammatical and typographical errors that with no doubt have to be corrected. There are, indeed, sentences where the phrasing is so confusing that the reader may come to false conclusions. In the current form, the paper is not acceptable for publication.

Thank you for your comment. We are very grateful for all of your suggestions and comments on the revision of this manuscript. The mistakes in the text have been corrected, and the manuscript was checked by an English editor.

Units in the whole manuscript should be standardized. For example, once there is K (line 81), another time °C (line 189).

The temperature unit was standardized.

I believe that in line 36 there should be “MPa“ instead of “mPa".

The sentence was corrected.

In line 127 there should be “cyclohexanone” instead of “cyclohexane”.

The word “cyclohexane” was changed to “cyclohexanone” in the sentence.

In line 186 in “Eg” letter g should be written with a lower index.

The letter g was written with a lower index.

In lines 203-204 it is said that the ranges of water concentration and amount of catalyst are 2226-2666 mmol/L and 4.88-6.16 g/L, respectively. Meanwhile, in table 1 these ranges are greater (for example experiments 2 and 7). Why?

The values in table 1 are approximate. To avoid confusion, 1 decimal place was added to the catalyst load values.

The figures are wrongly numbered and consequently wrongly mentioned in the text. Additionally, there are two “Figures 3” in the manuscript.

All figure numbers in the manuscript were corrected.

Reviewer 2 Report

The manuscript by Contreras and coworkers describe the ue of Degussa P-25 as potential photocatalyst for visible light oxidation of cyclohexane to cyclohexanol and cyclohexanone. This is a very important C-H activation reaction that suffers from low yields in general. Thus, this work is towards the correct direction. Overall, the work si well performed and presented. MInor corrections are required. In more detail:

Page 3, line 83: for the determination not for determines.

Page 3, line 85; Figure 3 should be figure 2.

3. Page 4, figure 3: should be figure 2.

4. In the same figure, what happens if the authors employ different solvents? Since MeCN is better than neat conditions, the authors should try other solvents as well.

5.Page 4, line 116, figure 2A and 2B should be figure 3A and 3B.

page 5, figure 2, should be figure 3

7 figure 3 should be figure 4. please change appropriate text.

figure 4 should be figure 5, change the text appropriately.

9. Page 9 reference 4, also add: Org. Biomol. Chem., 2018,16, 4596.

Author Response

The manuscript by Contreras and coworkers describe the use of Degussa P-25 as potential photocatalyst for visible light oxidation of cyclohexane to cyclohexanol and cyclohexanone. This is a very important C-H activation reaction that suffers from low yields in general. Thus, this work is towards the correct direction. Overall, the work is well performed and presented. Minor corrections are required. In more detail:

Thank you for your comment. We are very grateful for all your suggestions and comments in the revision of this manuscript.

1.  Page 3, line 83: for the determination not for determines.

“For determines” was changed to “to determine” in the sentence. The manuscript was also reviewed by an English editor.

2.  Page 3, line 85; Figure 3 should be figure 2.

All figure numbers in the manuscript were corrected.

3.  Page 4, figure 3: should be figure 2.

All figure numbers in the manuscript were corrected.

4.  In the same figure, what happens if the authors employ different solvents? Since MeCN is better than neat conditions, the authors should try other solvents as well.

Although we did not try another solvent in the present study, we agree that it would be very interesting to evaluate the photocatalytic oxofunctionalization of cyclohexane in different solvents. We will take it into account in a future study.

5.  Page 4, line 116, figure 2A and 2B should be figures 3A and 3B.

All figure numbers in the manuscript were corrected. 

6.  page 5, figure 2, should be figure 3

All figure numbers in the manuscript were corrected.

7.  figure 3 should be figure 4. please change appropriate text.

All figure numbers in the manuscript were corrected.

8. figure 4 should be figure 5, change the text appropriately.

All figure numbers in the manuscript were corrected.

9.  Page 9 reference 4, also add: Org. Biomol. Chem., 2018,16, 4596.

The mentioned reference was added to the bibliography.

Reviewer 3 Report

The authors report the results of the cyclohexanol/cyclohexanone conversion from cyclohexane using a commercial TiO2 powder as a catalyst. The key observation is that photocatalytic cyclohexane conversion under visible light is optimized using MeCN as the inert solvent. However, as the general reader, also working in the field of photocatalysis and catalyst materials, I’m attaching some comments and suggestions to the authors that have come up to my mind after reading the manuscript and some references:

It is not clear for me if there is a real improvement of these oxidation reactions.  The room temperature photocatalytic oxidation of cyclohexane over TiO2 has been studied more than 20 years ago (see for example reference 18). It is rather intuitive that conversion under VIS-light is going to be less than UV conditions and the mechanism of the reactions has also been extensively reported. The study concerning the amount of TiO2 is rather flat and does no add valuable information to the reader.

Regrettably, my recommendation is to reject this manuscript in its present form. My advice to authors is to test the properties of new materials or TiO2 using different morphologies.  

Author Response

The authors report the results of the cyclohexanol/cyclohexanone conversion from cyclohexane using a commercial TiO2 powder as a catalyst. The key observation is that photocatalytic cyclohexane conversion under visible light is optimized using MeCN as the inert solvent. However, as the general reader, also working in the field of photocatalysis and catalyst materials, I’m attaching some comments and suggestions to the authors that have come up to my mind after reading the manuscript and some references:

It is not clear for me if there is a real improvement of these oxidation reactions. The room temperature photocatalytic oxidation of cyclohexane over TiO2 has been studied more than 20 years ago (see for example reference 18). It is rather intuitive that conversion under VIS-light is going to be less than UV conditions and the mechanism of the reactions has also been extensively reported. The study concerning the amount of TiO2 is rather flat and does not add valuable information to the reader.

Regrettably, my recommendation is to reject this manuscript in its present form. My advice to authors is to test the properties of new materials or TiO2 using different morphologies.

We appreciate your analysis and the critique of our manuscript. Our work is focused in the development of greenchemistry applications. In this context, we chose the oxofunctionalization of cyclohexane as it is a well-known model reaction system of organic transformation. Also, in this work, we used direct solar light irradiation as a green energy source for the conversion of cyclohexane to cyclohexanol and cyclohexanone. This manuscript is focused on the use of an experimental design in order to find the optimal conditions for the highest conversion of cyclohexane. In a recent study, we evaluated the effect of exposing the facets of bismuth oxyiodide on the photocatalytic oxidation of cyclohexane. In this study we found that the ratio (001/110) play an important role in the selectivity of cyclohexane oxofunctionalization reaction. In another study, we evaluate the photocatalytic performance of N-TiO2and Fe-TiOmaterials. However, these were not better than Degussa for cyclohexane oxofunctionalization.

Round 2

Reviewer 1 Report

In my opinion paper can be accepted for publication in its current form.

Author Response

Thank you very much for your help

Reviewer 3 Report

The new document is clearly more readable. I’m still thinking that the results obtained from the authors don’t represent an improvement of the photocatalytic reaction. It is not author’s fault since it is well knowing that titania band-gap can be overcome using UV conditions and solar UV radiation is a relatively small part of the spectrum. This drawback can be solved by using heterojunctions. It allows one tuning the band gap of the composite avoiding the recombination of radicals (see for example Adv. Mater. 2017, 1601694). The authors said that “the manuscript is focused on the use of an experimental design in order to find the optimal conditions for the highest conversion of cyclohexane”. If the experimental design is the main objective of the article a more detail description of the solar experiments and the solar-reactor will be needed. Pictures of the solar collector are welcomed as supplementary information.

Afterwards, the paper could be accepted for publication.

Author Response

The main achievement of this manuscript is the optimal conditions determined by a multivariated strategy. In this way, a response polynomial and response surfaces (Figures 3 (a) and 3 (b)) were obtained.  In this way, the maximal performance of Degussa P25 TiO2for oxofunctionalization of cyclohexane on acetonitrile was reached. We agree with the TiOdoes not work appropriately under solar radiation. However, this is one of the cheapest (and available) photocatalyst. In this way we explore the maximal performance of this on simulated solar radiation. The optimal reaction conditions were validated under real solar radiation. In addition, in the manuscript the text: “A PAR radiometer, sensitive between 400 nm and 600 nm (QSPL-2100) was keeping inside reactors arrangement. A picture of this reactor is showed in S1” was added between lines 211-213, and a supplementary figure was added with the photograph of the reactors arrangement. Between lines 246-247: the text “The reactor picture is showed in figure S2” and a supplementary figure was added with the picture of this system.